| Open Peer Review | Microbial Ecology | Methods and Protocols

# MVP: a modular viromics pipeline to identify, filter, cluster, annotate, and bin viruses from metagenomes

Clément Coclet,[1] Antonio Pedro Camargo,[1] Simon Roux[1]

**ABSTRACT**    While numerous computational frameworks and workflows are available for recovering prokaryote and eukaryote genomes from metagenome data, only a limited number of pipelines are designed specifically for viromics analysis. With many viromics tools developed in the last few years alone, it can be challenging for scientists with limited bioinformatics experience to easily recover, evaluate quality, annotate genes, dereplicate, assign taxonomy, and calculate relative abundance and coverage of viral genomes using state-of-the-art methods and standards. Here, we describe Modular Viromics Pipeline (MVP) v.1.0, a user-friendly pipeline written in Python and providing a simple framework to perform standard viromics analyses. MVP combines multiple tools to enable viral genome identification, characterization of genome quality, filtering, clustering, taxonomic and functional annotation, genome binning, and comprehensive summaries of results that can be used for downstream ecological analyses. Overall, MVP provides a standardized and reproducible pipeline for both extensive and robust characterization of viruses from large-scale sequencing data including metagenomes, metatranscriptomes, viromes, and isolate genomes. As a typical use case, we show how the entire MVP pipeline can be applied to a set of 20 metagenomes from wetland sediments using only 10 modules executed via command lines, leading to the identification of 11,656 viral contigs and 8,145 viral operational taxonomic units (vOTUs) displaying a clear beta-diversity pattern. Further, acting as a dynamic wrapper, MVP is designed to continuously incorporate updates and integrate new tools, ensuring its ongoing relevance in the rapidly evolving field of viromics. MVP is available at https://gitlab.com/ccoclet/mvp and as versioned packages in PyPi and Conda.

**IMPORTANCE**    The significance of our work lies in the development of Modular Viromics Pipeline (MVP), an integrated and user-friendly pipeline tailored exclusively for viromics analyses. MVP stands out due to its modular design, which ensures easy installation, execution, and integration of new tools and databases. By combining state-of-the-art tools such as geNomad and CheckV, MVP provides high-quality viral genome recovery and taxonomy and host assignment, and functional annotation, addressing the limitations of existing pipelines. MVP's ability to handle diverse sample types, including environmental, human microbiome, and plant-associated samples, makes it a versatile tool for the broader microbiome research community. By standardizing the analysis process and providing easily interpretable results, MVP enables researchers to perform comprehensive studies of viral communities, significantly advancing our understanding of viral ecology and its impact on various ecosystems.

**KEYWORDS**    viromics pipeline, sequencing data, phages, viruses, ecological studies

The rapid expansion of sequencing technologies has provided a large amount of valuable data for mining uncultivated viral diversity from metagenomic/viromic assemblies that have greatly increased the number of virus genomes in public databases

Address correspondence to Clément Coclet, ccoclet@lbl.gov.

The authors declare no conflict of interest.

*[This article was published on 1 October 2024 with errors in references 4 and 5. The References were corrected in the current version, posted on 11 October 2024.]*

(1, 2). For instance, Integrated Microbial Genomes (IMG)/Virus (VR) currently provides access to a large collection of >5 millions viral sequences obtained from (meta)genomes, including both DNA and RNA viruses, either identified as viral contigs or integrated proviruses in genomes. Similarly, multiple studies, for example, *Tara* Oceans (3–5), and the human gut microbiomes (6–8), have performed metagenomics across ecosystems, collectively leading to the detailed characterization of the global diversity of DNA viruses and their abundance patterns on local and global scales (9, 10). For other ecosystems such as soils, the diversity and roles of viruses are poorly constrained, mostly due to the high complexity of these microbiomes (11). Viral-fraction metagenomes (viromes) have been highlighted as a promising approach to expand known viral diversity (12, 13). Notably, in 2014, a combined assembly of multiple viromes resulted in the discovery of the most abundant and widespread phage in the human gut, called crAssphage (14). Metatranscriptomics has also been a recent and powerful approach used for both viral activity measurement (15), and RNA virus discovery, that have uncovered tens of thousands of new uncultivated RNA viruses (16–18). Finally, recent metagenomic studies revealed important characteristics of environmental viral communities. For instance, in addition to their significant contribution to biogeochemical cycles through the lysis of their bacterial hosts, bacteriophages may also affect the diversity and function of marine microbial populations through the incorporation and expression of a broad range of auxiliary metabolic genes (AMGs) (19), and the number and functional diversity of these potential AMGs has rapidly increased through careful analysis of (viral) metagenomes (10, 20, 21).

Over the last decade, viromics analyses, meant here as the analysis of viral genomes from metagenomes, viromes, and/or metatranscriptomes, have coalesced around a number of core standard "steps" performed in the vast majority of studies. The first and most critical step is the computational identification of viral genomic sequences in metagenome assemblies, which relies on the use of sequence classification models as currently implemented in VirSorter2 (22), VIBRANT (23), and/or geNomad (24). Next, multiple tools are specifically dedicated to the analysis of these metagenome-derived virus genomes, including CheckV for genome completion and quality estimates (25), vRhyme for virus genome binning, CoverM for calculating coverage by read mapping, iPHoP for predicting hosts of viruses (26), or DRAM-v for functional annotation of viral contigs (27). Beyond these tools and approaches, multiple curated virus databases such as NCBI RefSeq (28), VOGDB (29), and IMG/VR (2) can guide virus taxonomic classification and functional annotation. Across published studies, these different tools and databases are either used individually or within large and complex workflows required for comprehensive analyses of viral diversity and ecology. As such, understanding which tools to use, how to integrate and connect different methods, and how to handle and interpret results is often challenging for users with limited familiarity with viruses and/or bioinformatic skills. Integrated pipelines providing an entire workflow for viromics analyses with easy‑to‑read results can significantly advance the field of viromics and contribute to democratize the study of viruses from sequencing data.

Some integrated pipelines have been developed in the last few years, such as MetaPhage (30), Viral Eukaryotic Bacterial Archaeal (VEBA) (31), ViWrap (32), Soil Virome Analysis Pipeline (SOVAP) (33), Multi-Domain Genome Recovery (MuDoGeR) (34), and ViromeFlowX (35), each proposing distinct features for the exploration of viromics data (Table 1). MetaPhage, MuDoGeR, ViWrap, and ViromeFlowX are modular pipelines that act as wrappers for several tools to study viruses from sequencing data. These pipelines integrate alignment-free VirFinder (36) and/or DeepVirFinder (37), and marker-based VIBRANT (23) and VirSorter2 (22) tools to identify and annotate viruses. The SOVAP and the VEBA use the hybrid method geNomad (24) to extract viral sequences from sequencing data. All pipelines, except SOVAP, assess the quality and remove low confidence viral predictions, using CheckV (25). All pipelines provide also a virus clustering step, using either dRep (38), Cluster Database at High Identity with Tolerance (CD-HIT) (39), vConTACT2 (40), or FastANI (41), and integrate tools for estimating the

**TABLE 1** MVP's features compared to other currently available viromics pipeline[a]. HMM: Hidden Markov Model; DRAM: Distilled and Refined Annotation of Metabolism.

| | MetaPhage (October 2022) | Veba2 (March 2024) | ViWrap (May 2023) | Sovap (May 2023) | MuDoGer (November 2023) | ViromeFlowX (February 2024) | MVP |
|---|---|---|---|---|---|---|---|
| Viral identification | DeepVirFinder Phigaro VIBRANT VirFinder VirSorter2 | VirFinder geNomad | VIBRANT VirSorter2 DeepVirFinder | geNomad | VIBRANT VirSorter2 VirFinder | VirSorter2 VirFinder | geNomad |
| Quality/completeness | CheckV | CheckV | CheckV | - | CheckV | CheckV | CheckV |
| Virus clustering | CD-HIT | FastANI | vConTACT2 dRep | CD-HIT | gOTUpick | CD-HIT | Blast-based greedy clustering (provided by CheckV) |
| Read mapping | Bowtie2 BamToCov | Bowtie2 Samtools SeqKit | CoverM | Samtools | Bowtie2 | Bowtie2 CoverM BEDTools | Bowtie2 (short reads) Samtools Minimap (long reads) CoverM |
| Functional annotation (databases) | DIAMOND | UniRef50 MIBiG VFDB CAZy Pfam KOFAM | KEGG | NCBI | - | GO EGGNOG KEGG PfamA EC CAZy | PHROGS Pfam dbAPIS RdRP HMM profiles DRAM-v pre-processing[b] |
| Taxonomic annotation | vConTACT2 | geNomad | RefSeq VOG | geNomad | vConTACT2 | RefSeq | geNomad |
| Binning | - | - | vRhyme | - | - | - | vRhyme |
| Preparation of metadata (MIUViG) for database submission | - | - | - | - | - | - | Yes |

[a]Blank cells indicate that the feature is not explicitly mentioned in the pipeline workflow.
[b]HMM: hidden Markov model; DRAM: distilled and refined annotation of metabolism.

abundance of recovered viral contigs and creating coverage tables. Finally, MetaPhage, SOVAP, MuDoGer, ViWrap, and ViromeFlowX integrate additional modules or analyses including taxonomic assignment, functional annotation, or host prediction, using a different set of tools. ViWrap in particular is the only pipeline at this time that includes viral binning, using vRhyme, in its workflow. Each pipeline has its unique strengths and features; however, all come with certain limitations. Some of these pipelines are not exclusively designed for viromics and instead have a broader focus on all microbial populations, which can lead to sub-optimal analysis results given the specificity of viral genome analyses. For example, the use of databases that are not virus-specific can lead to low-level or inaccurate functional annotations. Additionally, several of these pipelines have not been updated to use the latest generation of tools for viral detection, limiting their efficiency. Flexibility in handling input sequencing data or using intermediary outputs along the pipeline can also be a constraint in certain cases. Lastly, some pipelines lack documentation and generate output data that may not be user-friendly or easily interpretable, posing challenges in understanding and downstream utilization.

To address these limitations, we developed Modular Viromics Pipeline (MVP; an integrated and user-friendly pipeline designed exclusively for viromics analyses. MVP is currently organized into 10 modules and designed to be easily installed and executed, making it accessible for a wide range of users, even those who are new to bioinformatics and command-line environments. MVP combines geNomad, the most robust tool for viral genome recovery to date, with CheckV to assess the quality and filter the retrieved viral contigs. It also integrates several recent approaches including an automated filtering step, a robust handling of provirus sequences, that is, sequences including both a viral and host regions, virus-specific functional annotation, and a standardized pipeline to easily generate abundance matrices across a set of metagenomes. Through

each step, MVP generates easily readable result files along with overview summaries of the results. By providing an additional resource for researchers to perform viromics analyses, especially to address viral ecology and evolution questions, MVP will enable more microbiome researchers to study viruses in their sequencing data, expanding our collective understanding of their genetic diversity, distribution, function, evolution, and impacts across ecosystems.

## MATERIALS AND METHODS

The version of MVP described in this publication is MVP v1.0. MVP can be installed in multiple ways to accommodate different user preferences and system environments. The source code of MVP is available on a public repository (https://gitlab.com/ccoclet/mvp), allowing users to download and install it directly from the source. Additionally, MVP is packaged as a Conda package (MViP), facilitating easy installation and management of dependencies through the Conda package manager. MVP was primarily developed using Python programming language, leveraging various Python modules and libraries for different functionalities. Some of the key Python modules used in MVP include argparse for parsing command-line arguments, subprocess for executing shell commands, os for interacting with the operating system, pandas v.2.0.3 for data manipulation and analysis, and Biopython v.1.83. MVP is currently divided into ten modules: one set-up module (Module 00), seven analysis modules (Module 01–07), one metadata preparation module for genome database submission (Module 99), and one final module that summarizes all outputs generated along MVP pipeline (Module 100) (Fig. 1). Each module in MVP generates a summary report, which provides a comprehensive overview of the executed tasks, any errors encountered, and relevant output files. Furthermore, to maintain consistency and ease of use, the command-line interface of MVP follows a standardized pattern, with flags for specifying the working directory, metadata file, force option for overwriting existing files, sample group designation, and thread allocation for parallel processing. This uniformity ensures clarity and simplicity in executing MVP commands across different modules.

Before running MVP, users must first set up a metadata file and prepare a folder (or folders) containing the assembly files and the corresponding read files, as input files for MVP. Assembly files can be obtained from any sequencing data types (i.e., metagenomics, metatranscriptomics, viromics, single-cell amplified sequences). It is important to note that MVP does not offer a module for the pre-processing of the raw sequences (quality control [QC] control and assembly steps). The metadata file must list the paths of the assemblies and read sequence files to be processed, along with associated sample information (i.e., sample name and group). First, Module 00 ensures that the input data meet the necessary prerequisites, sets up the directory structure, and optionally installs databases, if the --install-databases flag is provided, for the subsequent analyses using the MVP pipeline. Specifically, the script checks the metadata file, as well as the input files to ensure their availability and validity. These preparatory steps ensure that MVP can effectively process and analyze the provided data.

Module 01 uses assembly files as the input source for geNomad v1.7.6 to identify viruses and proviruses. CheckV v1.0.1 is used on the outputs of the geNomad analysis (sequence files of predicted viral contigs, i.e., sample_name_virus.fna) to estimate the qualities and completeness of the recovered genomes. CheckV returns FASTA files containing sequences of both predicted viral and proviral contigs, that is, virus.fna and provirus.fna as well as a report table quality_summary.tsv that contains integrated results from CheckV. If CheckV identifies additional provirus sequences (provirus.fna not empty), that is, geNomad predictions that seemingly still included a host region, MVP automatically runs a second round of geNomad and CheckV specifically on these trimmed provirus sequences. This allows for the proper processing of proviruses trimmed by CheckV and makes sure the geNomad score and CheckV metrics associated with these sequences are based only on the trimmed region and not on a host region. By default,

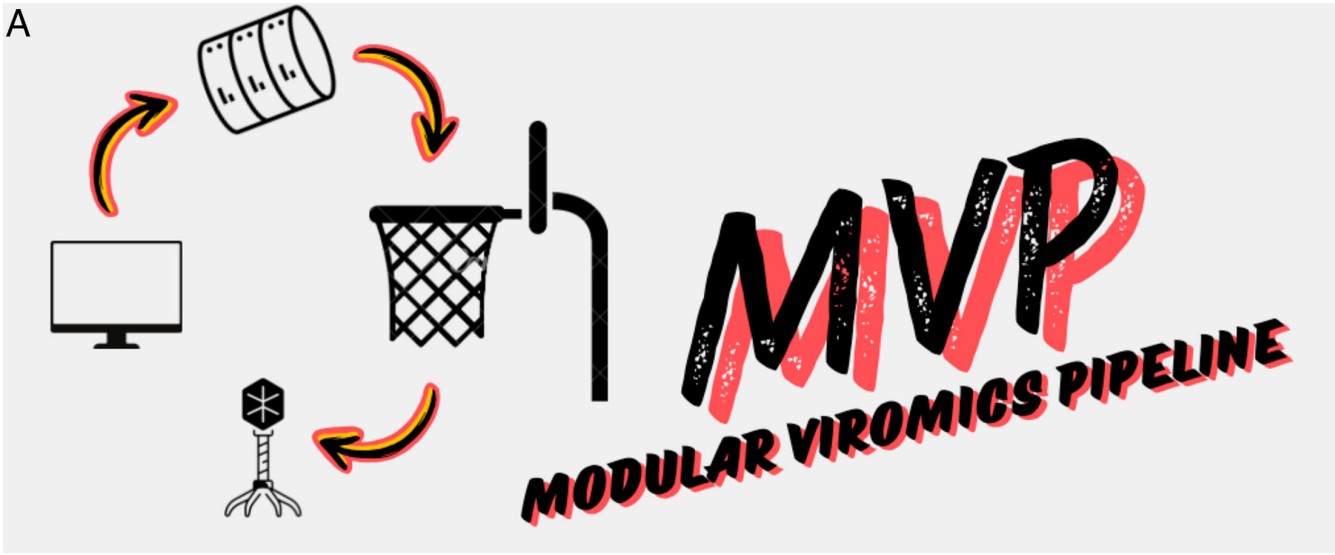

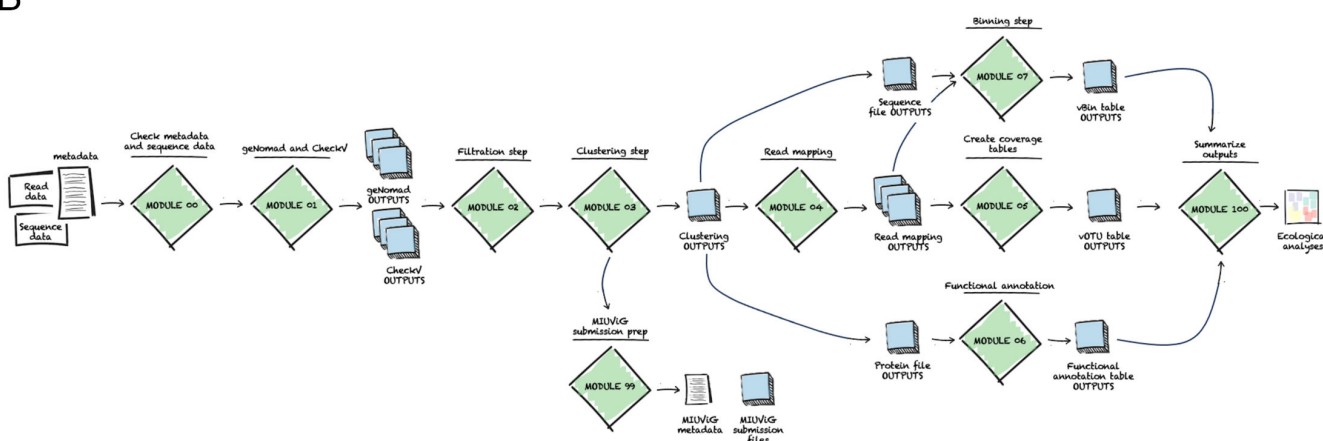

FIG 1   MVP logo and workflow describing the different steps and functionalities. MVP pipeline is divided in 10 modules: one set-up module (Module 00), seven analysis modules (Module 01–07), one preparation module for NCBI submission (Module 99), and one final module that summarizes all outputs generated along MVP pipeline (Module 100). White charts indicate inputs (assembly, read files, and a metadata), green diamonds indicate the modules that contain third-part tools and Python language to process inputs and generate outputs (blue squares).

MVP applies a conservative filtration based on post-classification filters (i.e., virus score ≥ 0.8, genome length ≥2,500 bp, and ≥1 virus hallmark genes detected by geNomad), preventing sequences without strong support from being classified as virus. To disable the conservative post-classification filters, the --genomad-relaxed flag can be added to the command, in which case, these cutoffs are changed to virus score ≥ 0.7, genome length ≥2,500 bp, and ≥0 virus hallmark genes. The same filtration parameter (i.e., --genomad-conservative or --genomad-relaxed) is used for both the initial and the second rounds of geNomad and CheckV. Module 01 also includes options for customization, including --modify-headers (default: TRUE) which appends each sample name as a prefix to the headers of the corresponding assembled sequences, and --min-seq-size, which enables the filtering of assembly sequences to be processed by geNomad based on size. These flags can be used to either mitigate potential errors due to identical sequence names across assemblies or to reduce the processing time of Module 01 by reducing the size of input files. Finally, Module 01 includes custom functions to merge both viral and proviral sequences into a FASTA file and create the corresponding report table. In particular, Module 01 includes custom functions to associate each viral sequence to a predicted genome type (dsDNA, ssDNA, RNA, etc.) and putative host domain

(prokaryotic vs. eukaryotic) based on its low-level taxonomic affiliation. For instance, all sequences identified as belonging to the *Caudoviricetes* class are associated with a "dsDNA"-predicted genome type and "prokaryotic"-predicted host group.

Module 02 performs a post-processing of the geNomad and CheckV outputs, and saves the results into filtered tables and FASTA files, for further analysis in subsequent modules. The subset files are based on two flags --viral-min-genes and --host-viral-genes-ratio, which enable the filtering of viral sequences based on the number of viral genes (default: 1) and the ratio between the host and viral genes (default: 1). This module is separated from Module 01 to enable a user to easily apply different cutoffs on these two parameters (number of viral genes, ratio between host and viral genes) without reprocessing the sequences with geNomad and CheckV.

Module 03 performs a vOTU-level clustering on filtered viral sequences previously identified across all samples listed in the metadata file. The default parameters for clustering are an average nucleotide identity (ANI) > 95% (--min_ani ≥ 0.95) and an alignment fraction (AF) > 85% (--min_tcov ≥ 0.85). The AF refers to the coverage of the shorter genome. Module 03 uses blast v2.14.1 and two custom python scripts to generate a table with the representative viral contigs, the membership contigs for each cluster with the information about each representative viral contigs along with all information derived from geNomad and CheckV. The ANI is calculated using the anicalc.py script, which processes the BLASTN results. Specifically, the script combines local alignments between sequence pairs to compute a global ANI by taking the average of nucleotide identities across all aligned regions between the query and the target sequences. Then, the aniclust.py script performs a greedy clustering based on the calculated ANI and the AF. The representative viral contig or bin for each vOTU is selected as the longest sequence from each cluster. Module 03 generates a FASTA file containing sequences of all representative viral contigs, which is utilized to construct an index used for Module 04 (read mapping). The index construction process in Module 03 employs either bowtie2 v2.5.3 or minimap2 v2.26, determined by the sequencing technology specified using the --read-type argument, that is, short-read or long-read sequencing, respectively. Finally, Module 06 uses four files for functional annotation: two FASTA files containing predicted protein sequences and two functional annotation tables. These files, produced by geNomad, cover both representative viral contigs and all viral contigs.

Module 04 aligns short- or long-sequencing reads against the index of representative virus sequences provided in Module 03 using Bowtie2 v2.5.3 or minimap2 v2.26, respectively. This alignment step may be processed on single paired-end read files (default: --interleaved TRUE), single unpaired read files (--interleaved FALSE), or paired R1/R2 read files, and returns sequence alignment/map (SAM) alignment files. The SAM alignment files are converted and sorted to produce BAM files using Samtools v1.19.2. A coverage table for each sample is then generated by CoverM v0.7.0. Reads are filtered using the classic filtration thresholds inherent to Bowtie2 and Minimap2, ensuring high-confidence alignments. Mapped reads are further filtered using a custom script based on horizontal coverage, performed in Module 05.

Module 05 summarizes results generated along the MVP pipeline (i.e., contig features from geNomad and CheckV, and coverage from read mapping step) and returns a set of coverage tables, performing additional filtration including standard cutoffs in viromics analyses applied to horizontal coverage (42) (default: --covered-fraction 0.1, 0.5, 0.9). By default, MVP applies a conservative filtration, selecting only viral sequences longer than 5 kb or longer than 1 kb and either complete, high-, or medium-quality for inclusion in the final coverage table. If the --filtration relaxed option is used, tables only undergo a filtering similar to that in Module 02 (i.e., number of viral genes and ratio between the host and viral genes).

Module 06 utilizes the FASTA file containing predicted protein sequences to perform the functional annotation of predicted viral proteins for either representative viral contigs or all viral contigs (default: --fasta-files representative). The protein sequences are derived from geNomad and predicted using pyrodigal-gv. First, Module 06 uses

the MMseqs2 v14.7e284 "search" workflow with a high sensitivity (-s 7) to compare all sequences in the protein FASTA file with all profiles in the PHROGS v.14 (43) and Pfam v37.0 (44) databases. Optionally, Module 06 provides the capability to compare protein sequences with a viral anti-prokaryotic immune system (APIS) protein database (dbAPIS) (45) (--anti-defense system [ADS] option) and/or RdRP HMM profiles (46) (--RdRP option), using blastp v2.14.1 and/or HMMER v3.4 hmmsearch program. After each annotation search, two tables are generated: an unfiltered one with all hits and a filtered one, in which hits are filtered based on standard scores and E-value cutoffs adjusted for each database when needed (Table S1). All the results obtained are then combined together along with the gene annotation table generated by geNomad into a single table. Finally, Module 06 includes custom functions to generate input files for DRAM-v v1.4.1 annotation (--DRAM option), in case users want to identify potential AMGs in their data set.

Module 07 is an optional module that let users perform a viral genome-binning step by vRhyme v1.1.0, using the viral contigs and the sorted BAM files produced by the Module 01 and Module 04, respectively, as inputs. The predicted vBin sequences are then used as input to CheckV to estimate the qualities and completeness of the binned viral genomes. Because of the fact that CheckV requires a single‐scaffold virus as an input at this point, multiple‐scaffold viral bins are concatenated with 10 Ns as linkers to meet the requirement. A read mapping, similar to that in Module 04, is performed, utilizing the same steps and providing identical options (i.e., --read-type and --interleaved). The best vBins are selected based on cutoffs recommended by vRhyme, and that vBins undergo either conservative (default) or relaxed filtration modes. In the conservative mode, MVP retains all vBins with less than two protein redundancy, guided by the observation that bins with approximately 2–5 redundant proteins may not be contaminated, albeit there are few such examples. Conversely, the relaxed mode only filters out vBins with more than five protein redundancy, as bins with >6 redundant proteins are often contaminated. Notable exceptions include nucleocytoplasmic large DNA viruses (NCLDVs), which can have ~10 redundant proteins in an uncontaminated bin. Summarized results from all modules, including unbinned contigs, vBins, geNomad, and CheckV features, and coverage results are then combined into tables, that can be used for downstream analyses. Finally, Module 07 includes custom functions to generate input files for iPHoP v1.3 (26), in case users want to computationally predict the host taxonomy from viral genomes.

Module 99 is another optional module intended to assist users submitting selected metagenome-assembled viral genomes to a public database such as NCBI GenBank. In a first step, this module gathers the necessary information from the previous modules (e.g., number of predicted coding sequences (CDS), geNomad score, estimated quality by CheckV, etc.) based on the contig identifier provided by the user. This first step then generates an intermediary file for the user to review and complete with metadata that cannot be obtained from previous MVP modules, such as environment type, sample location, and so on. After completing and reviewing this file, the user can execute the second step of this module, which verifies that all information is available and then uses table2asn v1.28 (47) to generate gbf and sqn files that can be used for GenBank submission. The format and metadata requirements and conventions are currently based on the latest published guidelines for releasing metagenome-assembled viral genomes (1, 48), and will be updated when new or updated guidelines are established.

Finally, Module 100 is an optional module that creates a summary report containing all the MVP commands used, the total running time, and a summary of the main results. The module organizes the main outputs tables in a folder to facilitate downstream analyses. Additionally, Module 100 includes R scripts to generate overview figures.

We illustrate the use of the MVP pipeline by processing a data set of 20 deeply-sequenced metagenome libraries, originally generated from sediment samples collected in the Loxahatchee Nature Preserve in the Florida Everglades (49, 50) (Fig. S1). Five samples (biological replicates) were collected at four different locations (Lox South, Lox

West, Lox North, and Lox East), resulting in 20 metagenome samples (Fig. S1). These libraries can be found in the IMG/M system (51) and have bgeen processed by the DOE Joint Genome Institute (JGI) Metagenome Workflow, an integrated workflow that includes read filtering, read error correction and assembly, structural and functional annotation of assembled contigs, and prokaryotic genome binning (52) (Table S2).

## RESULTS

### Folder structure of the MVP pipeline

The resulting folders and output files are arranged in the working directory in the following order:

- 01_GENOMAD/
  - SAMPLE_NAME/
    - SAMPLE_NAME_Viruses_Genomad_Output/
    - SAMPLE_NAME_Proviruses_Genomad_Output/
    - MVP_01_Sample_name_Summary_Report.txt
- 02_CHECK_V/
  - SAMPLE_NAME/
    - SAMPLE_NAME_Viruses_CheckV_Output/
    - SAMPLE_NAME_Proviruses_ CheckV_Output/
    - MVP_01_Sample_name_Unfiltered_Virus_Provirus_geNomad_CheckV_Table.tsv
    - MVP_01_Sample_name_Unfiltered_Virus_Provirus_Sequences.fna
    - …
    - MVP_02_Sample_name_Filtered_Virus_Provirus_geNomad_CheckV_Table.tsv
    - MVP_02_Sample_name_Filtered_Virus_Provirus_Sequences.fna
    - MVP_02_Sample_name_Summary_Report.txt

The two main folders, 01_GENOMAD and 02_CHECK_V, contain the results of Module 01 and 02. This includes geNomad and CheckV runs on virus and provirus sequences, with each processed sample in a separate folder. Additionally, the combined results of geNomad and CheckV are provided, including an unfiltered table and a FASTA file per sample.

The 02_CHECK_V folder also contains results generated by Module 02. These include a filtered table, a FASTA file, which represent the filtered versions of the ones generated by Module 01, based on the chosen filtration mode (conservative or relaxed). Finally, a summary report containing the command line with the different arguments used is generated for each step.

- 03_CLUSTERING/
  - TMP/
  - MVP_03_All_Samples_Unfiltered_Virus_Provirus_geNomad_CheckV_Table.tsv
  - MVP_03_All_Samples_Filtered_Virus_Provirus_geNomad_CheckV_Table.tsv
  - MVP_03_All_Samples_Filtered_Virus_Provirus_Sequences.fna
  - MVP_03_All_Samples_Filtered_Representative_Virus_Provirus_geNomad_CheckV_Table.tsv
  - MVP_03_All_Samples_Filtered_Representative_Virus_Provirus_Sequences.fna
  - MVP_03_Sample_name_Summary_Report.txt

The 03_CLUSTERING folder contains merged unfiltered and filtered tables, compiling the results of all samples processed through MVP. A merged FASTA file containing sequences of all predicted viruses is also provided. The directory contains also the

clustering results, including a vOTU-level table and a FASTA file containing only the vOTU representatives of species-level clusters. A summary report is generated, containing the command line with the different arguments used. The report also includes a summary of the number of viruses, before and after filtration, the number of vOTUs, as well as their various features such as genome length, genome quality, and taxonomy. Finally, the TMP folder contains all intermediary files generated by the clustering step and used to create final output tables, including the pairwise comparison table and the cluster memberships table.

- 04_READ_MAPPING/
  - Reference.*.bt2
  - SAMPLE_NAME/
    - Sample_name.sam
    - Sample_name.bam
    - Sample_name_sorted.bam
    - Sample_name_CoverM.tsv
    - MVP_04_Sample_name_Summary_Report.txt

The 04_READ_MAPPING folder contains the reference index built from the vOTU representatives from 03_CLUSTERING, to which sequencing reads will be aligned. For read mapping results, one folder for each sample contains the sorted BAM files, and coverage tables generated by CoverM. Intermediary SAM and BAM files can be deleted after running Module 04 if argument –delete-files is used.

- 05_VOTU_TABLES/
  - MVP_05_All_Samples_Filtered_Representative_Virus_Provirus_Coverage_Table.tsv
  - MVP_05_All_Samples_Filtered_Representative_Virus_Provirus_HC0.1_Coverage_Table.tsv
  - MVP_05_All_Samples_Filtered_Representative_Virus_Provirus_HC0.5_Coverage_Table.tsv
  - MVP_05_All_Samples_Filtered_Representative_Virus_Provirus_HC0.9_Coverage_Table.tsv
  - MVP_05_Summary_Report.txt

The 05_VOTU_TABLES folder contains four different coverage tables based on read mapping (Module 04). These tables summarize information for each representative vOTU, including geNomad and CheckV features, taxonomy, and coverage for each sample. Three of these tables are additionally filtered based on three horizontal coverage thresholds (i.e., 10%, 50%, and 90% by default). The coverage tables generated are designed to be immediately usable in standard software such as R for data manipulation, ecological analyses, and graphical display.

- 06_FUNCTIONAL_ANNOTATION/
  - MVP_06_All_Samples_Unfiltered_Virus_Provirus_Protein_Sequences.faa
  - MVP_06_All_Samples_Filtered_Representative_Virus_Provirus_Protein_Sequences.faa
  - MVP_06_All_Samples_Filtered_Virus_Provirus_geNomad_Annotation.tsv
  - MVP_06_All_Samples_Filtered_Representative_Virus_Provirus_geNomad_Annotation.tsv
  - MVP_06_All_Samples_Filtered_Representative_Virus_Provirus_All_Annotations.tsv
  - 06_RDRP_ANNOTATION/
    - MVP_06A_RdRP_Profile_Output.txt

- MVP_06A_RdRP_Profile_Tab.txt
- MVP_06B_Formatted_RdRP_Profile_Tab.tsv
- MVP_06C_Filtered_Formatted_RdRP_Profile_Tab.tsv
  - ○ 06_DRAM_V/
    - MVP_06_All_Samples_Filtered_Representative_Virus_Provirus_DRAMv_Annotation_Input.tsv
    - MVP_06_All_Samples_Filtered_Representative_Virus_Provirus_Sequences_DRAMv_Input.fa

The 06_FUNCTIONAL_ANNOTATION folder contains FASTA files of predicted protein sequences used as inputs by Module 06 to annotate viral proteins. It also includes the functional annotation table generated by geNomad in Module 01, which is combined with viral protein annotations performed against various databases, such as PHROGS, PFAM, and an optional anti-defense system database. If respective arguments are provided, two additional subfolders, 06_RDRP_ANNOTATION and 06_DRAM_V, may be created. These contain RdRP annotation tables which can be used to perform RdRP phylogeny analyses and two input files (a table and a FASTA file) compatible with DRAM-v, respectively.

- • 07_BINNING/
  - ○ 07A_vRHYME_OUTPUT
    - vRhyme_best_bins.summary.tsv
    - vRhyme_best_bins.
    - MVP_07A_Unfiltered_vBins_geNomad_CheckV_Table.tsv
    - vRhyme_best_bins_fasta/
      - ☐ vRhyme_bin_*. fasta
  - ○ 07B_vBINS_CHECKV/
    - MVP_07B_vBin_Sequences_CheckV_Input.fna
    - CheckV_quality_summary.tsv
  - ○ 07C_vBINS_READ_MAPPING/
    - SAMPLE_NAME/
      - ☐ Sample_name.sam
      - ☐ Sample_name.bam
      - ☐ Sample_name_sorted.bam
      - ☐ Sample_name_vBins_CoverM.tsv
    - MVP_07C_Unfiltered_vBins_geNomad_CheckV_Coverage_Table.tsv
  - ○ 07D_vBINS_vOTUS_TABLES/
    - MVP_07D_Filtered_vBins_geNomad_CheckV_Coverage_Table.tsv
    - MVP_07D_Filtered_vBins_Unbinned_vOTUs_geNomad_CheckV_Coverage_Table.tsv
    - MVP_07D_Filtered_vBins_Unbinned_vOTUs_geNomad_CheckV_HC0.1_Coverage_Table.tsv
    - MVP_07D_Filtered_vBins_Unbinned_vOTUs_geNomad_CheckV_HC0.5_Coverage_Table.tsv
    - MVP_07D_Filtered_vBins_Unbinned_vOTUs_geNomad_CheckV_HC0.9_Coverage_Table.tsv

The 07_BINNING folder contains the results of viral genome binning using vRhyme and related downstream analyses, resulting in four subfolders. Subfolder 07A_vRHYME_OUTPUT contains original vRhyme outputs, including two tables representing vBin membership information and FASTA files of best vBin sequences, along with a merged table summarizing vBin features (i.e., memberships, taxonomy, predicted hosts). Subfolder 07B_vBINS_CHECKV contains the merged FASTA file of the best vBin sequences, used as input for CheckV, and an output table representing vBin completeness information. Subfolders 07C_vBINS_READ_MAPPING

and 07D_vBINS_vOTUS_TABLES have similar hierarchies and contents to those in 04_READ_MAPPING and 05_VOTU_TABLES, respectively. The main difference is that coverage tables in 07D_vBINS_vOTUS_TABLES include information on both vBins and unbinned vOTUs.

- 99_GENBANK_SUBMISSION/
    - ○ UViG_metadata_tables/
        - ▪ contig_name_annotation.tsv
        - ▪ contig_name_metadata.tsv
    - ○ UViG_submission_files/
        - ▪ contig_name_genome.sqn
        - ▪ contig_name_genome.gb

The 99_GENBANK_SUBMISSION folder contains a metadata file generated by the first step of Module 99 that needs to be reviewed and completed to process the second step. Subfolder contains genbank (.gb) and .sqn files required for GenBank submission.

- 100_SUMMARIZED_OUTPUTS/
    - ○ DATE-TIME/
        - ▪ Date-time_MVP_100_Summary_Report.txt
        - ▪ MVP_*_Output_table.tsv
        - ▪ Summarize_Output_Plots.pdf

Finally, the 100_SUMMARIZED_OUTPUTS folder contains a summary report, which includes all MVP commands, the main outputs tables generated throughout the MVP pipeline, and a PDF file with multiple figures. These files are stored in a subfolder named by the date and time Module 100 is run, allowing users to execute it multiple times without overwriting previous files.

## MVP benchmarking using 20 metagenome samples

The metagenome of 20 sediment samples from 4 different locations (i.e., South, West, North, and East) in the Loxahatchee Nature Preserve was previously processed using the JGI Metagenome Workflow (52) (Table S2). The number of filtered reads per library ranged from 240 to 478 million, and the number of contigs ranged per library ranged from 2.87 to 7.22 million (Table S2). From these, 6 high-quality and 122 medium-quality genomes bins were recovered across the 20 metagenomic libraries (Table S2). Using a minimum geNomad score of 0.7, we predicted 21,037 putative viral contigs, including 346 proviruses, before filtration (Fig. 2A), ranging from 3.3 to 207 kb, with mostly low-quality or unknown quality genomes (99.4%) (Fig. S2A). After filtration (relaxed mode: minimum number of viral genes = 1; maximum ratio of host genes to viral genes = 1), 11,656 putative viral contigs, including 339 proviruses, were kept, ranging from 3.8 to 207 kb, with mostly low-quality or unknown quality genomes (98.9%) (Fig. S2B). After clustering genomes (ANI ≥ 95; aligned fraction [AF] ≥ 85), MVP recovered 8,298 "species-level" vOTUs, including 225 proviruses (Fig. 2B). This initial number includes all detected vOTUs before applying any specific filtration criteria. Among these, 1,437 "species-level" vOTUs, including 57 proviruses, were identified using the conservative filtration mode. This mode selects low-quality genomes larger than 5 kb or complete, high-, or medium-quality and larger than 1 kb. These criteria ensure that only high-confidence viral sequences are included in the final analysis (Fig. 2B and C). Regardless of filtration and dereplication, the number of predicted viruses at the South site was consistently lower than at the other sites, which may reflect variations in microbiome diversity and/or library quality between sites. A marker gene taxonomic classification performed using geNomad suggested that the vast majority of vOTUs belonged to the double-stranded DNA *Caudoviricetes* class (94.7%), while 4.5% remained unclassified. These tailed prokaryotic viruses represent the most abundant group of phages in most environments, and their dominance were expected in these libraries given that

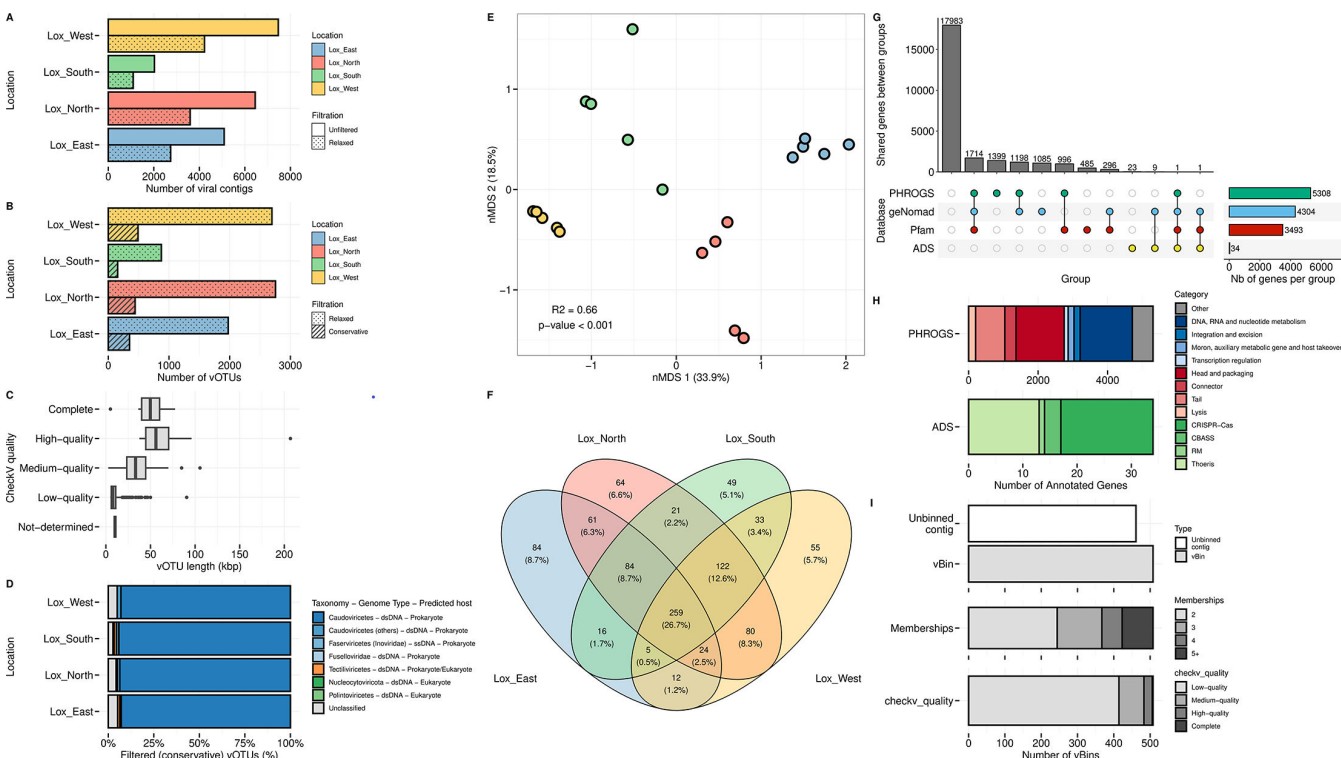

**FIG 2** Characterization of Viral Contigs and Viral Operational Taxonomic Units (vOTUs) across the 20 metagenome samples (4 locations) and quality assessments. (A) Distribution of viral contigs across four locations. The number of viral contigs is displayed for unfiltered data (plain) and relaxed filtration (dot). (B) Distribution of vOTUs (ANI ≥ 95; AF ≥ 85) across the locations. The number of vOTUs is shown for relaxed (dot) and conservative (stripe) filtration. (C) Quality assessment of vOTUs (after conservative filtration) using CheckV. The length of vOTUs (in kbp) is shown separately for each CheckV quality category: not-determined, low-quality, medium-quality, high-quality, and complete. (D) Taxonomic composition of filtered (conservative) vOTUs. The percentage of vOTUs in each location is categorized by taxonomy. This panel provides insight into genome type and predicted host of the viral communities. (E) Non-metric multidimensional scaling (nMDS) ordination plot showing beta-diversity of viral communities. The nMDS plot illustrates the differences in viral community composition among the four locations, with $R^2$ and $P$-values indicating the significance of the differences observed. (F) Venn diagram of shared vOTUs between locations. (G) Upset plot showing number of shared annotated genes between databases (ADS, Pfam, geNomad, PHROGS). (H) Proportion of annotated genes based on functional annotation against PHROGS (blue and red) and dbAPIS (green). Functional categories associated with lytic infections are colored in red, and the other major phage functional categories are colored in blue. (I) Distribution of viral bins (vBins) by vRhyme and unbinned representative vOTUs. The number of vBins is shown by CheckV quality (low-quality, medium-quality, high-quality, complete) and the number of representative vOTU memberships (2, 3, 4, 5+). This panel provides an overview of the viral binning analysis.

the majority of the microbial contigs and metagenomes-assembled genomes (MAGs) belonged to bacterial phyla (49).

After predicting, filtering, and dereplicating, the viral genomes from the 20 assemblies, a read mapping step is performed. This process involves mapping metagenomic reads onto the provided metagenomic assemblies or viromes to obtain scaffold coverage. Overall, 259 (26.7%) vOTUs were found at least in one sample of each location (Fig. 2F). Conversely, 252 (26.1%) vOTUs were only found in a specific location, with the East site exhibiting the highest number of unique vOTUs ($n = 84$; 8.7%). These patterns were confirmed by Bray–Curtis dissimilarity metric, non-metric multidimensional scaling (nMDS) analyses, showing that viral communities differed significantly by location (PERMANOVA test; $R^2 = 0.66$; $P$-value = 0.001), built based on the final coverage table generated by MVP step 05 (Fig. 2E). This pattern of significant clustering by location was consistent whether the data set was filtered by horizontal coverage or not (Fig. S3A through C), and using both vBins and unbinned contigs (Fig. S3D through H).

To explore the functional potential of these viruses, protein-coding genes were predicted and compared to the Pfam-A (44), TIGRFAM (53), KEGG Orthology (54)

and COG (55) databases by geNomad. In total, 8,645 (34.3%) genes were functionally annotated, with 9.20% of genes annotated by virus-specific markers. To provide additional information, the same predicted genes were also assigned to PHROGS (43) and dbAPIS (45) databases, resulting in the functional annotation of 5,309 (21.1%), and 1,399 (5.55%) predicted genes, respectively (Fig. 2G and H). Regarding counter-defense mechanism, most predictions were either CRISPR-Cas or Thoeris APISs.

Finally, 508 viral bins (vBins) were reconstructed from 8,298 representative vOTUs, using vRhyme. Most vBins were composed of either 2 ($n$ = 244) or 3 ($n$ = 123) members, while 7,441 viral contigs remained unbinned (Fig. 2I). Among these, vBin genomes ranged from 5 to 131 kb, with 94 of them being either complete, high- or medium-quality genomes (Fig. 2I).

The total running time for processing the 20 assemblies and generating all results presented above, from Module 00 to Module 100 was 229 h, 19 min, and 48 s on 64 CPUs. The most time-consuming parts took 212 h and 19 min (10 h and 36 min per assembly) to predict viral genomes and estimate their quality, using geNomad and CheckV, respectively. The second most time-consuming part took 17 h and 20 min (52 min per assembly) for the read mapping step.

## Comparison to ViWrap pipeline

To compare the performance of MVP to ViWrap v.1.3.0 (32), another modular pipeline that uses different virus identification tools (i.e., VIBRANT, and VirSorter2), we used a subset of the original metagenome libraries ($n$ = 8; two replicates per location), as the inputs (Fig. S4). The total running time for processing the eight assemblies and generating all results presented below (Fig. S4) was 99 h, 25 min, and 27 s (approximately 16 h, 88 min, and 58 s per assembly), representing a running time 1.5 longer per library compared to MVP. Using VIBRANT v.1.2.1 (23), with a minimum contig length of 5 kb, the number of predicted viral contigs ranged from 202 to 1,865, representing 4,868 viral contigs (Fig. S4A). After applying the same filtration thresholds (relaxed and conservative) used for MVP, the number of viral contigs ranged from 46 to 309 per location, showing a significant decrease mostly due to the removal of predicted viral contigs without any viral gene. After clustering, 4,562 viral genomes (vOTUs) were reconstructed, including both binned and unbinned viruses (Fig. S4B), indicating that most of vOTUs are singletons. The same decrease in number of vOTUs was observed as for viral contigs after both relaxed and conservative filtration, resulting in 864 and 862 vOTUs, respectively. The majority of filtered (conservative mode) vOTUs are low-quality genomes, while high-quality and complete vOTU genomes are relatively rare (Fig. S4C). Respectively, 17.0% and 14.8% were either taxonomically assigned or had a predicted host (Fig. S4D and E). Among these, and similarly to MVP analyses, the vast majority (95.0%) of the annotated vOTUs belonged to *Caudoviricetes* (Fig. S4D). Finally, the most common predicted bacterial hosts are *Desulfobacterota*, followed by *Mycobacteriales*, and *Alphaproteobacteria* (Fig. S4E).

## DISCUSSION

MVP is a modular and comprehensive pipeline that integrates cutting-edge tools and software for complete viral analysis from metagenomic data. Unlike previously developed pipelines, which typically focus on specific steps of virus analysis such as virus identification, taxonomic classification, or virus binning, MVP stands out for its capability to conduct end-to-end viromics analysis using the latest and most efficient tools. It is specifically designed to handle and combine results from large sets of metagenomes. Importantly, MVP reduces the burden on users to benchmark and choose suitable software and tools for their analyses. This standardized approach ensures MVP can consistently deliver reproducible results in a user-friendly manner. MVP generates summary reports at various steps of the viral analysis, which provide a quick overview of the commands used, as well as intermediary statistics of taxonomic annotation, genome quality estimation, and coverage.

MVP integrates numerous state-of-the-art, recent, and popular tools designed for viromics analysis, and uses a modular organization in which the inputs and outputs of each step are connected. MVP seamlessly runs with all the settings preconfigured, allowing users who may not want to explore custom options and parameters for each tool to obtain meaningful results for downstream analyses. MVP can process different types of data sets (metagenomes, metatranscriptomes, or viromes) or read inputs for mapping (paired or unpaired short or long reads). For more advanced users, MVP also offers the possibility to apply customized thresholds, allowing different levels of filtration, and the use of various databases for functional annotation. The pipeline also allows users to customize their analyses by skipping optional steps, such as read mapping or binning, and focusing on specific functionalities.

By comparing the two pipelines, MVP appears faster to run considering the same data set. The end-to-end MVP workflow allow multiple assemblies as inputs and will generate both single-assembly and combined-assemblies' outputs, which allow the users to compare results per assembly. ViWrap generates unfiltered summary tables containing predicted viral contigs without any viral gene, which may bias further analyses, while MVP provides filtered outputs that users can directly utilize. However, ViWrap also offers features and modules not yet available in MVP, such as host prediction or AMG annotation.

Although MVP application was tested here with samples from a natural environment (sediment samples from mangroves), the tools and databases implemented in MVP allow it to be widely used for all types of samples, such as human microbiome, wastewater or plant-associated microbiome samples, for example. With the rapid growth of the field of viral ecology, larger data sets and more advanced tools are being constantly developed and released. The modular nature of MVP will ensure easy integration of these new tools and databases for the future releases of MVP. We plan to collect user issues and suggestions through various channels, including GitLab for issue tracking and feature requests, as well as actively engaging with users through community forums, social media, and direct feedback mechanisms. Additionally, we will incorporate user feedback into the development of future versions of MVP to ensure continuous improvement and alignment with user needs and preferences. Some potential additions include creating a new module to integrate vConTACT3 (https://bitbucket.org/MAVERICLab/vcontact3/src/master/), the latest iteration in the vConTACT taxonomic classifiers, which is currently in beta version and actively being developed. Another additional feature considered is the integration of host prediction using the tool iPHoP (26). Integration of additional tools and/or databases will be prioritized based on user feedback provided, for example, through the ticket system associated with the MVP repository (https://gitlab.com/ccoclet/mvp). Given MVP's features and future improvements, MVP has the potential to be widely adopted by the microbiome research community, enabling standardized and comprehensive studies of viral diversity.

## ACKNOWLEDGMENTS

We thank Josué Rodríguez-Ramos for testing and providing feedback on MVP pipeline.

The work conducted by the U.S. Department of Energy Joint Genome Institute (https://ror.org/04xm1d337), a DOE Office of Science User Facility, is supported by the Office of Science of the U.S. Department of Energy operated under Contract No. DE-AC02-05CH11231. This work was supported by the U.S. Department of Energy, Office of Science, Biological and Environmental Research, Early Career Research Program awarded under UC-DOE Prime Contract DE-AC02-05CH11231.

## AUTHOR AFFILIATION

[1]DOE Joint Genome Institute, Lawrence Berkeley National Laboratory, Berkeley, California, USA

## AUTHOR ORCIDs

Clément Coclet  http://orcid.org/0000-0002-6672-148X

## AUTHOR CONTRIBUTIONS

Clément Coclet, Conceptualization, Investigation, Methodology, Software, Validation, Visualization, Writing – original draft, Writing – review and editing | Antonio Pedro Camargo, Methodology, Validation, Writing – review and editing | Simon Roux, Conceptualization, Investigation, supervision, Validation, Writing – original draft, Writing – review and editing

## ADDITIONAL FILES

The following material is available online.

### Supplemental Material

**Supplemental Figures (mSystems00888-24-s0001.docx).** Figures S1, S2, S3, and S4.
**Supplemental legends (mSystems00888-24-s0002.docx).** Legends for supplemental figures and tables.
**Supplemental Tables (mSystems00888-24-s0003.xlsx).** Tables S1, S2, and S3.

### Open Peer Review

**PEER REVIEW HISTORY (review-history.pdf).** An accounting of the reviewer comments and feedback.

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
