## [Reviewer comments · mSystems]

MVP: a Modular Viromics Pipeline to identify, filter, cluster, annotate, and bin viruses from metagenomes.

Clément Coclet, Antonio Camargo, and Simon Roux

Corresponding Author(s): Clément Coclet, DOE Joint Genome Institute

Review Timeline:

Submission Date:	July 1, 2024
Editorial Decision:	July 31, 2024
Revision Received:	August 9, 2024
Accepted:	September 9, 2024

Editor: Alejandro Reyes Munoz

Reviewer(s): Disclosure of reviewer identity is with reference to reviewer comments included in decision letter(s). The following individuals involved in review of your submission have agreed to reveal their identity: Cristina Moraru (Reviewer #1)

Transaction Report:

DOI: <https://doi.org/10.1128/msystems.00888-24>

Re: mSystems00888-24 (MVP: a Modular Viromics Pipeline to identify, filter, cluster, annotate, and bin viruses from metagenomes.)

Dear Dr. Clément Coclet:

Two reviewers have revised the manuscript and both find it interesting and relevant. Reviewer #1 requests several clarifications and improvements on the document, which I consider relevant and important but shouldn't affect the overall message or impact of the manuscript, reason why I'm considering those as minor modifications.

Revision Guidelines

Sincerely,
Alejandro Reyes Munoz
Editor
mSystems

Reviewer #1 (Comments for the Author):

The MVP tool looks interesting and useful. It comes with a corresponding github page, containing installation and usage instructions, and from where MVP can be downloaded. However, the manuscript needs to be significantly improved.

Main points to address for manuscript improvement: i) inconsistencies in the text and figure 1B in the number of modules and their description, ii) missing info regarding the metagenomes used for benchmarking; iii) unstructured Results section; iv) it would be nice to have line numbers for the next review round. These points are described in detail below.

There is a certain inconsistency in the number of modules and their description in between the last paragraph in the Introduction, Materials and methods and Figure 1B. In the introduction and legend for Fig. 1B, it is stated that MVP consists of 10 modules. However, in Materials and methods there are only eight modules identified (01 to 08). In Fig. 1B, there are depicted 10 modules, however, their numbering is not contiguous: module 08 is missing, and then there are mentioned modules 99 and modules 100. Likely, module 99 from Fig. 1B corresponds with module 08 from Materials and methods, however, the keyword "MIUViG" is missing from the description of module 08. Furthermore, I could not figure out where in Materials and methods is module 100 (from Fig. 1B) described. Please address all the inconsistencies in module description and numbering between the different sections of the manuscript.

The metagenomes used for benchmarking are insufficiently described. First, there is no corresponding paragraph in Materials and methods, where at least their label and IMG/M accession number is given. Second, it would be nice to shortly mention from which habitat type do these metagenomes originate (river, soil, etc.), so that the reader doesn't need to go searching in the cited literature for this information. Third, the text mentions 20 metagenomes, however, in Fig. 2 only four labels appear (Lox South, Lox West, Lox North and Lox East). I could not find in the text / figure legend how the 20 metagenomes relate to the four labels. At this point it is unclear how many metagenomes were analyzed.

The Results section should be structured using subheadings, in at least two parts: one describing the MVP outputs and the other describing the benchmarking results.

MVP-related points:

- Module 3

o you are using "aligned fraction" as threshold for vOTU clustering. Please mention whose "aligned fraction": of the query, of both the query and subject?

o describe how the "average nucleotide identity" was calculated from the blast results. I assume blastn was used?

o Describe how the representative viral contigs for each vOTU are selected

- Module 4:

o Are mapped reads filtered in any way, more specifically based on their percent identity to the reference?

Benchmarking-related points:

- When describing the number of vOTUs obtained in the Results, it is unclear what the "1437" vOTUs represent.

- In Fig. 2, B and C: do the statistics here correspond to representatives viral contigs from each vOTU?

- Viral bins: where only the representative contigs for each vOTU used for binning, or all viral contigs in each vOTU?

- In Fig. 2, there is no panel J or K (as indicated in the Results section, when reporting on viral bins).

Reviewer #2 (Comments for the Author):

The manuscript "MVP: a Modular Viromics Pipeline to identify, filter, cluster, annotate, and bin viruses from metagenomes." from Coclet et al., introduces MVP, an end-to-end pipeline for viromics analysis.

I initially attempted to install MVP using conda but the environment had still not resolved within 24 hours. I was then able to install with mamba quite rapidly. I suspect the conda issue may be me using a slightly older version of conda. In any case, I was easily able to install it and run the test dataset.

I found the documentation to be clear and thorough. This is always greatly appreciated, especially as it will increase the accessibility for those with less commandline experience.

Regarding the tool itself, MVP is effectively a wrapper of the cutting-edge tools that are forming the gold standard of viromics analyses. The modules are clearly well thought out and I have very little to add in this respect. I like MVP and think it will be of much use to the community.

I also enjoyed the associated manuscript. I think the short form suits this type of work well and I enjoyed the emphasis on the different modules.

Overall, I think this is a very good tool and a good writeup that will be of interest to the community. Kudos to the authors! I have only a small number of minor comments/queries.

- Although it is now very much out of date, I think it may be appropriate to acknowledge MetaPhage as this pipeline tried to address a similar need to MVP (<https://doi.org/10.1128%2Fmsystems.00741-22>)

- Near the end of Methods, there are two sequential sentences beginning as such "Finally, Module 07" and "Finally, module 08". I'd modify one of the two.

- Have the author's considered including Phables for the resolution of phage assembly graphs? Emerging data seems to suggest that it works well (<https://doi.org/10.1093/bioinformatics/btad586>)
- I see that MVP currently clusters vOTUs using BLAST with the CheckV method. Do the authors have any thoughts on this new pre-print for clustering vOTUs? (<https://www.biorxiv.org/content/10.1101/2024.06.27.601020v1.full>). Not a suggestion, I'm genuinely just curious...

Reviewer #1 (Comments for the Author)

The MVP tool looks interesting and useful. It comes with a corresponding GitHub page, containing installation and usage instructions, and from where MVP can be downloaded. However, the manuscript needs to be significantly improved. Main points to address for manuscript improvement: i) inconsistencies in the text and figure 1B in the number of modules and their description, ii) missing info regarding the metagenomes used for benchmarking; iii) unstructured Results section; iv) it would be nice to have line numbers for the next review round. These points are described in detail below.

Thank you for your valuable feedback. We appreciate the thorough review and suggestions, and made the necessary modifications to enhance the quality of our work and manuscript. **We have added line numbers in the main manuscript to facilitate the next review round.**

1. There is a certain inconsistency in the number of modules and their description in between the last paragraph in the Introduction, Materials and methods and Figure 1B. In the introduction and legend for Fig. 1B, it is stated that MVP consists of 10 modules. However, in Materials and methods there are only eight modules identified (01 to 08). In Fig. 1B, there are depicted 10 modules, however, their numbering is not contiguous: module 08 is missing, and then there are mentioned modules 99 and modules 100. Likely, module 99 from Fig. 1B corresponds with module 08 from Materials and methods, however, the keyword "MIUViG" is missing from the description of module 08. Furthermore, I could not figure out where in Materials and methods is module 100 (from Fig. 1B) described. Please address all the inconsistencies in module description and numbering between the different sections of the manuscript.

Thank you for pointing out the inconsistencies in the module numbers and descriptions. We have resolved these inconsistencies in the main manuscript, Figure 1B, and the figure legend. To clarify:

- MVP consists of 10 modules in total.
- Modules 00 to 07 represent the main modules for viromics analyses.
- Module 99 (MIUViG preparation) is an optional module and is singular as it is not part of the linear workflow (only applies to individual virus genomes, not entire datasets).
- Module 100 is a module that summarizes all outputs along the MVP workflow.

We decided to use Modules 99 and 100 because MVP is under active development, and it is likely that more modules will be developed and added to the general workflow in the future.

We listed below all the modifications we made in the main manuscript:

- We replace “7 command lines” with “using only 10 modules executed via command lines” (1.XX).
- We clarify the number of modules, and their purposes along MVP pipeline in the Methods section: “MVP is currently divided into ten modules: one set-up module (Module 00), seven analysis modules (Module 01 to Module 07), one metadata preparation module for genome database submission (Module 99), and one final module that summarizes all outputs generated along MVP pipeline (Module 100) (Figure 1).” (1.XX)
- We replaced “Finally, Module 08” with ‘Module 99’ (1.XX)
- We added a description for the Module 100: “Finally, Module 100 is an optional module that creates a summary report containing all the MVP commands used, the total running time, and a summary of the main results. The module organizes the main outputs tables in a folder to facilitate

downstream analyses. Additionally, Module 100 includes R scripts to generate overview figures.” (l.XX)

- We added folder structures for both Module 99 and Module 100, as well as explanations of subfolders and files: “
- 99_GENBANK_SUBMISSION/
 - UViG_metadata_tables/
 - contig_name_annotation.tsv
 - contig_name_metadata.tsv
 - UViG_submission_files/
 - contig_name_genome.sqn
 - contig_name_genome.gb

The 99_GENBANK_SUBMISSION folder contains a metadata file generated by the first step of Module 99 that needs to be reviewed and completed to process the second step. Subfolder contains genbank (.gb) and .sqn files required for GenBank submission.

- 100_SUMMARIZED_OUTPUTS/
 - DATE-TIME/
 - Date-time_MVP_100_Summary_Report.txt
 - MVP_*_Output_table.tsv
 - Summarize_Output_Plots.pdf

Finally, the 100_SUMMARIZED_OUTPUTS folder contains a summary report, which includes all MVP commands, the main outputs tables generated throughout the MVP pipeline, and a PDF file with multiple figures. These files are stored in a subfolder named by the date and time Module 100 is run, allowing users to execute it multiple times without overwriting previous files.” (l.XX)

- We replaced “Module 00 to Module 08” with “Module 00 to Module 100”. (l.XX)
- We also added the description of Modules (00 to 100) in the Fig. 1B caption: “MVP pipeline is divided in 10 modules: one set-up module (Module 00), seven analysis modules (Module 01 to Module 07), one genome metadata preparation module (Module 99), and one final module that summarizes all outputs generated along MVP pipeline (Module 100).”

2. The metagenomes used for benchmarking are insufficiently described. First, there is no corresponding paragraph in Materials and methods, where at least their label and IMG/M accession number is given. Second, it would be nice to shortly mention from which habitat type do these metagenomes originate (river, soil, etc.), so that the reader doesn't need to go searching in the cited literature for this information Third, the text mentions 20 metagenomes, however, in Fig. 2 only four labels appear (Lox South, Lox West, Lox North and Lox East). I could not find in the text / figure legend how the 20 metagenomes relate to the four labels. At this point it is unclear how many metagenomes were analyzed.

We thank the reviewer for raising this point. To better describe the metagenome samples used for MVP benchmarking, we have created a supplementary figure (Figure S1) and table (Table S2) that fully details these 20 samples (IMG/M accession number, number of reads, number of contigs, size, location, habitat, and general features). For better clarity, we have moved the first paragraph of the Results section into the Methods section and referenced the new supplementary table.

To clarify, we used 20 metagenome samples from sediments in the Loxahatchee Nature Preserve in the Florida Everglades. Five samples (biological replicates) were collected at four different locations (Lox South, Lox West, Lox North, and Lox East), resulting in 20 metagenome samples. In Figure 2, some panels show values (sum) between replicates for each location (panels A, B, D, F), while panel E shows the 20 metagenome samples colored by location.

We listed below all the modifications we made in the main manuscript:

- *“We illustrate the use of the MVP pipeline by processing a dataset of 20 deeply-sequenced metagenome libraries, originally generated from sediment samples collected in the Loxahatchee Nature Preserve in the Florida Everglades[48,49] (Figure S1). Five samples (biological replicates) were collected at four different locations (Lox South, Lox West, Lox North, and Lox East), resulting in 20 metagenome samples (Figure S1). These libraries can be found in the IMG/M system[50] and have been processed by the DOE Joint Genome Institute (JGI) Metagenome Workflow, an integrated workflow that includes read filtering, read error correction and assembly, structural and functional annotation of assembled contigs, and prokaryotic genome binning[51].” (1.XX)*
 - We added the new supplementary table and information regarding the number of samples: *“The metagenome of 20 sediment samples from 4 different locations (i.e., South, West, North, and East) in the Loxahatchee Nature Preserve was previously processed using the JGI Metagenome Workflow[51] (Table S2).” (1.XX)*
 - - We modified the Figure 2 caption to make it clearer: *“Figure 2. Characterization of Viral Contigs and Viral Operational Taxonomic Units (vOTUs) across the 20 metagenome samples (4 locations) and Quality Assessments.”*
3. The Results section should be structured using subheadings, in at least two parts: one describing the MVP outputs and the other describing the benchmarking results.

We thank the reviewer for this suggestion. We added 3 subheadings for the Results section:

- *Folder structure of the MVP pipeline (1.XX)*
 - *MVP benchmarking using 20 metagenome samples (1.XX)*
 - *Comparison to ViWrap pipeline (1.XX)*
4. MVP-related points
- 4a. Module 3:
- you are using "aligned fraction" as threshold for vOTU clustering. Please mention whose "aligned fraction": of the query, of both the query and subject?

We thank the reviewer for this question. We modified the main manuscript to make it clearer: *“The default parameters for clustering are an average nucleotide identity (ANI) greater than 95% (--min_ani >= 0.95) and an alignment fraction (AF) greater than 85% (--min_tcov >= 0.85). The alignment fraction refers to the coverage of the shorter genome.” (1.XX)*

- describe how the "average nucleotide identity" was calculated from the blast results. I assume blastn was used?

We added more details on how ANI is calculated in the main manuscript: “*The ANI is calculated using the anicalc.py script, which processes the BLASTN results. Specifically, the script combines local alignments between sequence pairs to compute a global ANI by taking the average of nucleotide identities across all aligned regions between the query and the target sequences.*” (1.XX)

- Describe how the representative viral contigs for each vOTU are selected

Thank you for the suggestion. We added explanation on how the representative contigs are selected: “*Then, the aniclust.py script performs a greedy clustering based on the calculated ANI and the alignment fraction (AF). The representative viral contig or bin for each vOTU is selected as the longest sequence in each cluster.*” (1.XX)

4b. Module 4:

Are mapped reads filtered in any way, more specifically based on their percent identity to the reference?

We thank the reviewer for the question. Mapped reads are filtered in Module 05 based on horizontal coverage. We clarified it in the main manuscript: “*Thank you for the suggestion. We added explanation on how the representative contigs are selected: “Then, the aniclust.py script performs CD-HIT-like clustering based on the calculated ANI and the alignment fraction (AF). The representative viral contigs for each vOTU are selected based on the sequence that has the highest coverage and similarity within the cluster, ensuring that the most representative sequence of the viral population is chosen.”*” (1.XX)

5. Benchmarking-related points

- When describing the number of vOTUs obtained in the Results, it is unclear what the "1437" vOTUs represent.

We thank the referee for noticing this issue. We identified 8,298 vOTUs after the clustering step. After applying the conservative filtration, the remaining number of vOTUs was 1,437. We clarified this in the main manuscript: “*After clustering genomes (Average Nucleotide Identity, ANI ≥ 95 ; Aligned Fraction, AF ≥ 85), MVP recovered 8,298 ‘species-level’ vOTUs, including 225 proviruses (Figure 2B). This initial number includes all detected vOTUs before applying any specific filtration criteria. Among these, 1,437 ‘species-level’ vOTUs, including 57 proviruses, were identified using the conservative filtration mode. This mode selects genomes larger than 5 kb or complete, high-, or medium-quality and larger than 1 kb. These criteria ensure that only high-confidence viral sequences are included in the final analysis (Figure 2B and 2C).*” (1.XX)

- In Fig. 2, B and C: do the statistics here correspond to representatives viral contigs from each vOTU?

Figure 2B shows statistics for vOTUs after both relaxed and conservative filtration modes. Figure 2C shows statistics for conservative vOTUs. We clarified this in the Figure 2 caption: “**(C)** *Quality assessment of vOTUs (after conservative filtration) using CheckV. The length of vOTUs (in kbp) is shown separately for CheckV quality category: not-determined, low-quality, medium-quality, high-quality, and complete.*”

- Viral bins: where only the representative contigs for each vOTU used for binning, or all viral contigs in each vOTU?

The reviewer is correct. Only representative vOTUs have been used for the binning step. We modified the main manuscript: “*Finally, 508 viral bins (vBins) were reconstructed from 8,298 representative vOTUs, using vRhyme.*” (1.XX) and in Figure 2 caption: “**(I)** *Distribution of viral bins (vBins) by vRhyme and unbinned representative vOTUs. The number of vBins is shown by CheckV quality (low-quality, medium-quality, high-quality, complete) and the number of representative vOTU memberships (2, 3, 4, 5+).*”

- In Fig. 2, there is no panel J or K (as indicated in the Results section, when reporting on viral bins).

Thank you for noticing this error: “*Most vBins were composed of either 2 (n = 244) or 3 (n = 123) members, while 7441 viral contigs remained unbinned (Figure 2I). Among these, vBin genomes ranged from 5 to 131 kb, with 94 of them being either complete, high- or medium-quality genomes (Figure 2I).*” (1.XX)

Reviewer #2 (Comments for the Author):

The manuscript "MVP: a Modular Viromics Pipeline to identify, filter, cluster, annotate, and bin viruses from metagenomes." from Coclet et al., introduces MVP, an end-to-end pipeline for viromics analysis. I initially attempted to install MVP using conda but the environment had still not resolved within 24 hours. I was then able to install with mamba quite rapidly. I suspect the conda issue may be me using a slightly older version of conda. In any case, I was easily able to install it and run the test dataset. I found the documentation to be clear and thorough. This is always greatly appreciated, especially as it will increase the accessibility for those with less command line experience. Regarding the tool itself, MVP is effectively a wrapper of the cutting-edge tools that are forming the gold standard of viromics analyses. The modules are clearly well thought out and I have very little to add in this respect. I like MVP and think it will be of much use to the community. I also enjoyed the associated manuscript. I think the short form suits this type of work well and I enjoyed the emphasis on the different modules. Overall, I think this is a very good tool and a good writeup that will be of interest to the community. Kudos to the authors! I have only a small number of minor comments/queries.

Thank you for your review and positive feedback on our manuscript and for taking the time to test MVP.

Regarding the issue with installing MVP using Conda, we experienced similar problems with the conda installation sometimes unable to resolve the mvip package, and we recommend using Mamba for a quicker and more reliable installation process. We updated our documentation to highlight this recommendation and help users avoid similar issues in the future.

1. Although it is now very much out of date, I think it may be appropriate to acknowledge MetaPhage as this pipeline tried to address a similar need to MVP (<https://doi.org/10.1128%2FmSystems.00741-22>)

Thank you for the suggestion. We added a column in Table 1 that presents MetaPhage’s features. We also modified the main manuscript:

- “*Some integrated pipelines have been developed in the last few years, such as MetaPhage[30], VEBA[31], ViWrap[32], SOVAP[33], Multi-Domain Genome Recovery (MuDoGeR)[34], and*

- ViromeFlowX*[35], each proposing distinct features for exploration of viromics data (Table 1). *MetaPhage*, *MuDoGeR*, *ViWrap* and *ViromeFlowX* are modular pipelines that act as wrappers for several tools to study viruses from sequencing data. These pipelines integrate alignment-free *VirFinder*[36] and/or *DeepVirFinder*[37], and marker-based *VIBRANT*[23] and *VirSorter2*[22] tools to identify and annotate viruses.” (1.XX)
- “Finally, *MetaPhage*, *SOVAP*, *MuDoGer*, *ViWrap*, and *ViromeFlowX* integrate additional modules or analyses including taxonomic assignment, functional annotation, or host prediction, using a different set of tools.” (1.XX)

2. Near the end of Methods, there are two sequential sentences beginning as such "Finally, Module 07" and "Finally, module 08". I'd modify one of the two.

Done: “Module 99 is another optional module intended to assist users submitting selected metagenome-assembled viral genomes to a public database such as NCBI GenBank.” (1.XX) and “Finally, Module 100 is an optional module that creates a summary report containing all the MVP commands used, the total running time, and a summary of the main results.” (1.XX)

3. Have the author's considered including Phables for the resolution of phage assembly graphs? Emerging data seems to suggest that it works well (<https://doi.org/10.1093/bioinformatics/btad586>)

We thank the reviewer for the suggestion. We agree that integrating Phables into the MVP pipeline may lead to better reconstruction of viral genomes, especially from complex and diverse metagenomic samples. Currently, MVP uses assemblies as inputs to identify viral contigs however, so a new module “Pre-processing” would need to be developed to integrate Phables. We plan to develop this module if our internal benchmark confirms that Phables substantially improve the recovery of viral genomes from metagenomes.

4. I see that MVP currently clusters vOTUs using BLAST with the CheckV method. Do the authors have any thoughts on this new pre-print for clustering vOTUs? (<https://www.biorxiv.org/content/10.1101/2024.06.27.601020v1.full>). Not a suggestion, I'm genuinely just curious...

We thank the reviewer for the suggestion. We already have a plan to replace the current approach for ANI calculation and clustering by vCust (<https://github.com/refresh-bio/vclust> and <https://www.biorxiv.org/content/10.1101/2024.06.27.601020v1>) in the next MVP version, which in our internal tests is faster and provides reliable results. For clustering, we are currently testing different approaches including one based on Leiden graph clustering.

Re: mSystems00888-24R1 (MVP: a Modular Viromics Pipeline to identify, filter, cluster, annotate, and bin viruses from metagenomes.)

Dear Dr. Clément Coclet:

Your manuscript has been accepted, and I am forwarding it to the ASM production staff for publication. Your paper will first be checked to make sure all elements meet the technical requirements. ASM staff will contact you if anything needs to be revised before copyediting and production can begin. Otherwise, you will be notified when your proofs are ready to be viewed.

Sincerely,
Alejandro Reyes Munoz
Editor
mSystems

Reviewer #2 (Comments for the Author):

The authors have fully and thoughtfully addressed all comments from myself and the other reviewer. I have no further comments.